# Wide range of G6PD activities found among ethnic groups of the Chittagong Hill Tracts, Bangladesh

**Benedikt Ley**[1]*, **Mohammad Golam Kibria**[2], **Wasif Ali Khan**[2], **Sarah Auburn**[1,3,4], **Ching Swe Phru**[2], **Nusrat Jahan**[2], **Fatema Tuj Johora**[2], **Kamala Thriemer**[1], **Jenifar Quaiyum Ami**[2], **Mohammad Sharif Hossain**[2], **Ric N. Price**[1,3,4], **Cristian Koepfli**[5]*, **Mohammad Shafiul Alam**[2]

**1** Global and Tropical Health Division, Menzies School of Health Research and Charles Darwin University, Darwin, Australia, **2** Infectious Diseases Division, International Centre for Diarrheal Disease Research Bangladesh (icddr,b), Dhaka, Bangladesh, **3** Mahidol-Oxford Tropical Medicine Research Unit (MORU), Faculty of Tropical Medicine, Mahidol University, Bangkok, Thailand, **4** Centre for Tropical Medicine and Global Health, Nuffield Department of Clinical Medicine, University of Oxford, Oxford, United Kingdom, **5** Eck Institute for Global Health, Department of Biological Sciences, University of Notre Dame, Notre Dame, Indiana, United States of America

* benedikt.ley@menzies.edu.au (BL); ckoepfli@nd.edu (CK)

**Data Availability Statement:** The corresponding database is submitted together with the manuscript as supporting information.

## Abstract

The proportion of *Plasmodium vivax* malaria among all malarias is increasing worldwide. Treatment with 8-aminoquinolines remain the only radical cure. However, 8-aminoquinolines can cause severe hemolysis in glucose-6-phosphate dehydrogenase (G6PD) deficient patients. The population of the multi-ethnic Chittagong Hill Tracts (CHT) carry the highest malaria burden within Bangladesh. As in many countries the national treatment guidelines recommend 8-aminoquinoline based radical cure without routine G6PD deficiency (G6PDd) testing to guide treatment. Aim of this study was to determine the need for routine testing within a multi-ethnic population by assessing the prevalence of G6PDd among the local population. Participants from 11 ethnicities were randomly selected and malaria status was assessed by microscopy, rapid diagnostic test (RDT) and polymerase chain reaction (PCR). G6PD status was determined by spectrophotometry and G6PD genotyping. The adjusted male median (AMM) was defined as 100% G6PD activity, participants were categorized as G6PD deficient (<30% activity), G6PD intermediate (30% to 70% activity) or G6PD normal (>70% activity). Median G6PD activities between ethnicities were compared and the association between G6PD activity and malaria status was assessed. 1002 participants were enrolled and tested for malaria. G6PD activity was measured by spectrophotometry in 999 participants and host G6PD genotyping undertaken in 323 participants. Seven participants (0.7%) had peripheral parasitaemia detected by microscopy or RDT and 42 by PCR (4.2%). Among 106 participants (32.8%) with confirmed genotype, 99 (93.4%) had the Mahidol variant. The AMM was 7.03U/gHb with 90 (9.0%) G6PD deficient participants and 133 (13.3%) with intermediate G6PD activity. Median G6PD activity differed significantly between ethnicities (p<0.001), proportions of G6PD deficient individuals ranged from 2% to 26% but did not differ between participants with and without malaria. The high G6PDd prevalence and

**Funding:** This study was funded by Bill and Melinda Gates Foundation (OPP1054404 and OPP1164105), and the Eck Institute for Global Health at the University of Notre Dame. BL is funded by the Australian Department of Foreign Affairs and Trade, RNP is funded by the Wellcome Trust (Senior Fellowship in Clinical Science, 200909), KT is a CSL Centenary Fellow. SA is supported by the Bill and Melinda Gates Foundation (OPP1054404), Wellcome Trust (200909 awarded to RNP) and a Georgina Sweet Award for Women in Quantitative Biomedical Science. The funders had no role in study design, data collection and analysis, decision to publish, or preparation of the manuscript.

**Competing interests:** The authors have declared that no competing interests exist.

significant variation between ethnicities suggest routine G6PDd testing to guide 8-aminoquinoline based radical in the CHT and comparable settings.

## Author summary

The *Plasmodium vivax* malaria parasite is a major public health burden in many parts of the Asia-Pacific and Americas. Primaquine-based radical cure is the only available treatment that effectively removes *P. vivax* from the human host but can cause severe side effects in individuals with glucose-6-phosphate dehydrogenase (G6PD) deficiency. The WHO recommends testing for G6PD deficiency to guide treatment, but this is not done on a routine basis in many countries. The aim of this study was to determine the importance of routine testing in multi-ethnic populations. The study was conducted in a multi-ethnic population in the Chittagong Hill Tracts (Bangladesh), which carries the highest burden of malaria within the country. We enrolled 1002 participants from 11 major ethnicities within the area and assessed the prevalence of G6PD deficiency by phenotype and genotype. Based on phenotype 9.0% of the population were not eligible for standard primaquine based radical cure, however numbers varied significantly between ethnicities, ranging from 2% to 26%. In 93% of cases where a genotype was identified, the severe Mahidol G6PD variant was found. Our findings highlight the importance of routine G6PD testing prior to primaquine based radical cure in multi-ethnic populations.

## Introduction

Great efforts in the reduction of malaria have been made over the last decade, but with a much greater impact on *Plasmodium falciparum (P. falciparum)* than the neglected *P. vivax* parasite; accordingly, the proportion of malaria cases caused by *P. vivax* is increasing worldwide [1]. In contrast to *P. falciparum*, *P. vivax* forms dormant liver stages (hypnozoites) that reactivate weeks to months after the first infection, causing significant morbidity and mortality [2]. Relapses arising from reactivation of hypnozoites are the main contributor to the transmission of *P. vivax* infections and may contribute to over 80% of clinical cases [3,4]. The 8-aminoquinolines primaquine (PQ) and tafenoquine (TQ) are the only licensed drugs that effectively remove hypnozoites from the human host and, while well tolerated in most recipients, can cause severe and potentially fatal hemolysis in patients with glucose-6-phosphate dehydrogenase deficiency (G6PDd) [5–12]. G6PDd is among the most common inherited enzymopathies worldwide with more than 400 million people affected; prevalence differs by ethnicity and is highest among populations at risk of malaria, possibly the result of a protective effect of G6PDd against some forms of malaria [13–15]. The WHO recommends routine testing for G6PDd to guide 8-aminoquinoline based radical cure [16,17], a recommendation frequently not followed since drug induced hemolysis is under-estimated, the impact of vivax malaria is not recognized and fears of additional costs to national public health systems [18,19].

Malaria has declined over the last decade in Bangladesh, with the population at risk falling to 17 million across 13 districts [20,21]. The highest rates of malaria are in the multi-ethnic Chittagong Hill Tracts Districts (CHT), on the eastern border with Myanmar, where a total of 20,446 clinical cases were reported in 2016 with an estimated incidence of 1.7 per 1,000 population.

*P. falciparum* remains the dominant malaria species in the country, however its proportion has dropped from 90% in 2007 to less than 80% in 2016, with a rising proportion of *P. vivax* cases [22–25]. As in many countries, PQ is provided for the radical cure of *P. vivax*, but patients are not tested for G6PDd to guide treatment and no systematic assessment of the prevalence of G6PDd within the country has been conducted. Anecdotal reports from the CHT estimate the G6PDd prevalence between 20% and 40% (Khan, personal communication), however in hospital based surveys in the same region only 0.6% to 1.4% of patients were G6PDd [25,26]. The significant differences in these estimates may reflect study populations with different ethnic backgrounds, and associated differences in *G6PD* genetic profiles, or an underlying difference in G6PD activity in patients with and without malaria [15,27]. Understanding whether the prevalence of G6PDd differs significantly between different ethnicities living in the same area is essential in informing decision-makers on whether and how to introduce routine testing for G6PDd into national malaria treatment guidelines. The aim of this study was to assess whether different ethnicities sharing the same risk of malaria infection would have significantly different prevalence of G6PDd.

## Methods

### Ethics statement

The study was approved by the Ethics Review Committee of the icddr,b, Bangladesh (PR-15021), the Human Research Ethics Committee of the Northern Territory Department of Health and Menzies School of Health Research, Australia (HREC 2015–2336), and the University of Notre Dame Institutional Review Board (18-09-4875). Written informed consent was collected from all participants or their legal guardians prior to enrolment and in addition written assent was collected from all minors above the age of 11 years.

### Study area

The CHT are densely forested, with limited accessibility and while the area covers approximately 10% of the country, it accounts for only 1% of the total population [28]. At least 12 major indigenous groups and a Bengali subpopulation are permanent residents of this region [29,30]. The CHT carry the highest burden of malaria within the country, incidence is seasonal with a major peak around June to August and a second smaller peak around February.

### Sampling

Most villages (paras) in the CHT are mono-ethnic [30]. To sample sufficient numbers of participants from all targeted ethnic groups, individual villages were selected purposefully according to the village size, accessibility and ethnicity. The average village comprises 46 households, a total of 34 villages and 2134 households were selected [30]. Each village was mapped using Google Maps (Alphabet Inc., Mountainview, USA) and individual households were identified, numbered and a subset selected at random.

To ensure a good representation of genetic differences in G6PD alleles, one person per household above the age of one year was selected randomly and invited to participate. Randomization was done using pre-generated randomization list from random.org [31] (last accessed 24.06.2020).

### Community sensitization, consent and sample collection

Selected villages were visited several days prior to participant enrolment to inform the population about the survey. Following written informed consent, a questionnaire and medical examination were completed including details on body weight, height and axillary body

temperature, and a self-reported medical history. A malaria rapid diagnostic test (RDT) (Falci-Vax, Zephyr Biomedicals, Goa, India), a malaria microscopy slide and hemoglobin (Hb) measurement (Hemocue 201, Angelholm, Sweden) were done on site. Participants were informed of the results of their malaria RDT and Hb concentration and those with a positive malaria RDT were referred to the closest medical facility for treatment. Participants above the age of 7 years were asked to contribute a venous blood sample (maximum of 5ml) which was collected in an EDTA tube (BD, Franklin Lakes, USA). In participants less than 7 years old a capillary sample (~400μl) was collected into an EDTA Microtainer (BD, Franklin Lakes, USA) tube.

## Sample processing

Slides were read at a local laboratory. Thick and thin films were stained with Giemsa and the parasite density quantified either per 500 white blood cells or 1000 red blood cells. All slides were read by two independent readers and the mean was recorded, provided that the discrepancy between readings was less than 50% of the smaller value and species identification was in accordance. The presence of gametocytes was recorded, but not quantified. Whenever discordant results were found the respective slides were shipped for reference reading to the International Centre for Diarrheal Diseases and Research, Bangladesh (icddr,b), Dhaka, and this result was considered as final.

Blood samples were stored immediately at 4°C and shipped under controlled conditions to the reference laboratory at the icddr,b in the capital Dhaka for quantitative G6PD measurements, host genotyping and *Plasmodium* species confirmation by qPCR. Since samples can hemolyze during transport, G6PD activity was only measured if there were no visible signs of hemolysis. G6PD measurement was done on a temperature controlled Shimadzu UV-1800 spectrophotometer (Kyoto, Japan), using kits and lyophilized controls from Randox Laboratories (Crumlin, UK), following procedures described previously, considering the Hb value measured in the field [32,33]. All samples were measured in duplicate, and a third measurement was performed if the measurements differed by more than 10% of the upper value. All procedures followed the manufacturer's instructions with normal and deficient controls run with every test run. The measurement temperature was set to 37°C, the absorption was read twice, five minutes apart, at 340 nm. G6PD activity was calculated from the difference in absorption between the two measurements, following a formula provided by the manufacturer; no temperature correction factor was applied. A run was repeated in case either control returned a result outside the range recommended by the manufacturer.

DNA from whole blood was extracted using QIAamp DNA Blood Minikit (Qiagen, Germany) according to manufacturer's instruction, and screened by qPCR for presence of malaria parasites. 4 μL of DNA, corresponding to 4 ml of blood, was screened by qPCR for *P. falciparum* using the varATS assay, and for *P. vivax* using the *cox1* assay. *P. falciparum* varATS qPCR was run in a total volume of 12 μL, containing 0.4 μM of each primer and probe, 6 μL TaqMan FastAdvanced (Applied Biosystems), and 4 μL DNA. *P. vivax* cox1 qPCR was run in a total volume of 12 μL, containing 0.4 μM of each primer and probe, 6 μL TaqMan FastAdvanced (Applied Biosystems), and 4 μL DNA. The varATS assay targets multicopy genes and amplifies approximately 20 copies per genome. Cox1 is a mitochondrial gene and present in ~10 copies per genome [34,35]. By qPCR, samples were only screened for *P. falciparum* and *P. vivax*. Samples were not screened for *P. malariae*, *P. ovale*, or *P. knowlesi*.

A subset of samples was genotyped for host G6PD variants known to be present in the study area (Mahidol, Viangchan, Mediterranean, Orissa and Kalyan-Kerala) by Sanger sequencing. PCR was performed in a final reaction volume of 20μL containing 10 x PCR buffer with MgCl$_2$ (25 mM), GC rich buffer, dNTP mixture (2.5 mM), forward (10 μM) and reverse

primers (10 μM), FastStart Taq DNA Polymerase (Roche, France) and DNA template. The primer sequences are provided in S1 Table.

Thermal cycling profile for all of the included variants except the Kalyan-Kerala variant were set following reference literature with slight modifications: pre-denaturation was done at 95˚C for 15 minutes, followed by 35 cycles of denaturation at 94˚C for 45 seconds, annealing at specific temperature for 30 seconds and extension at 72˚C for 1 minute 20 seconds, and a final extension at 72˚C for 10 minutes [36]. Thermal cycling profile for Kalyan-Kerala variant: pre-denaturation at 95˚C for 15 minutes, 35 cycles of denaturation at 94˚C for 35 seconds, annealing at 63˚C for 40 seconds and extension at 72˚C for 1 minute 10 seconds, and a final extension at 72˚C for 10 minutes. To yield better DNA bands, the annealing temperatures for Mahidol and Viangchan variants were set according to reference literature and were optimized for Mediterranean, Orissa and Kalyan-Kerala variants (S2 Table).

### Data management and statistical analysis

All participant and corresponding laboratory data were entered on paper forms, which were then digitalized using Epidata version 3.1 (Denmark). All analyses were done using Stata version 14 (College Station, USA).

G6PD activity, measured by spectrophotometry (in U/dL), was normalized by Hb measurement (in g/dL) to provide a result in U/gHb. The adjusted male median (AMM) was calculated and defined as 100% G6PD activity for the entire study population [37]. Individuals were categorized as being G6PD deficient if their enzyme activity was less than 30% of the AMM, G6PD intermediate if enzyme activity was between 30% to 70% activity, and G6PD normal if enzyme activity exceeded 70%. Age was assessed for normality and G6PD activity and age were then compared for significant differences between ethnic groups using the t-test, Mann—Whitney U test or Kruskal Wallis test as appropriate. Proportions of G6PD deficient, G6PD intermediate and G6PD normal participants per ethnic group were assessed for significant differences using a $\chi^2$-test or fishers exact test. Multiple regression analysis was performed to identify key variables affecting non-normalized G6PD activity. Non-normalized G6PD activity (in U/dL) was used as the dependent variable and Hb concentration included as a covariate [38]. The independent variables were backwards selected, initially considering age, gender, body weight, height and temperature, Hb, the delay between sample collection and processing, spectrophotometry result, ethnicity, and G6PD genotype. Participants were categorized by malaria status as being malaria positive by microscopy or RDT, positive by malaria PCR but not microscopy or RDT, or negative by all assays. Since categories showed strong dependency the model was repeated separately for microscopy/RDT and PCR diagnosis. Models were assessed for multi-co-linearity by calculating the variation inflation factor (VIF), with co-linearity defined when VIF was >10.

### Sample size

To detect a G6PDd prevalence of 5% with 4% absolute confidence limit and assuming erroneous procedures in 10%, would require a total of 100 individuals enrolled per ethnic group [39,40]. To achieve this a total of 10 distinct ethnicities were identified within the area that could be contacted with a total sample size of 1,000 participants [30]. Host genotyping was performed on all samples with less than 70% G6PD activity by spectrophotometry as well as 10 randomly selected samples per ethnicity from G6PD normal participants.

## Results

Between 13th August 2015 and 11th January 2016 a total of 1000 participants from 10 ethnic groups were enrolled into the study. In addition, two participants from

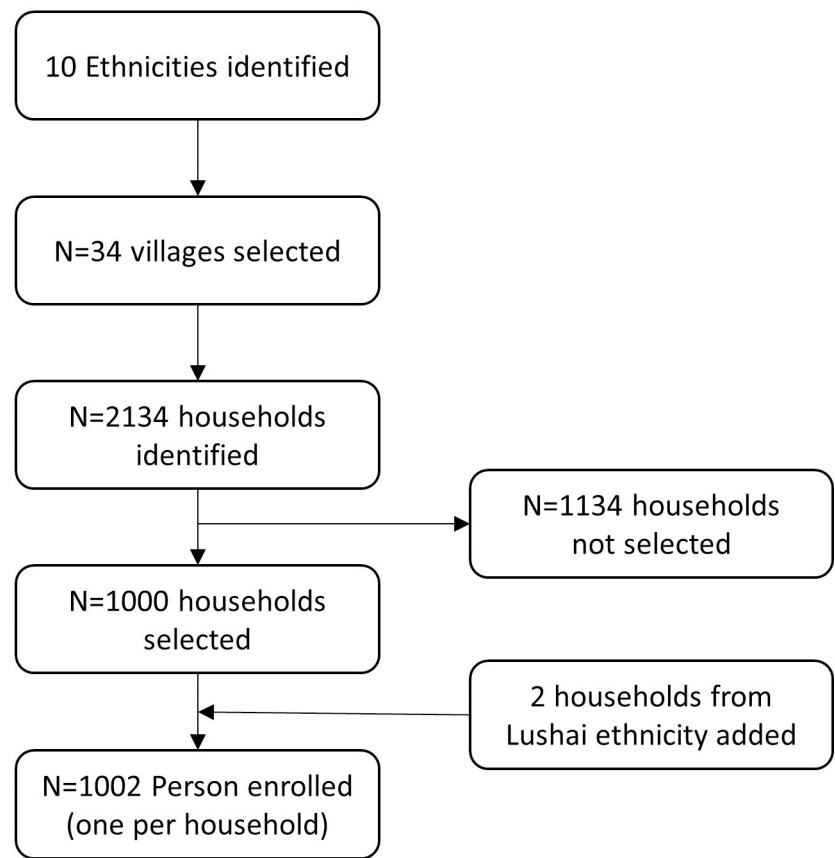

**Fig 1. Consort chart of participant selection process.**

the Lushai ethnic group were enrolled, bringing the total sample size to 1002 (Fig 1 and S1 File).

Spectrophotometry was not performed in three individuals, resulting in G6PD data from 999 (99.7%) participants (Table 1).

**Table 1. Overview on baseline data collected, stratified per ethnicity.**

| Ethnicity | Total | n Male (%) | n Female (%) | Mean Age in yrs. (95%CI) | Mean Hb in g/dL (95%CI) | Febrile (Temp >37.5C) n (%) | Mean body temp. in °C (95% CI) |
|---|---|---|---|---|---|---|---|
| Bawm | 100 | 43 (43.0) | 57 (57.0) | 39.9 (36.9–43.0) | 13.4 (13.1–13.7) | 1 (1.0) | 36.3 (36.3–36.4) |
| Bengali | 99 | 34 (34.3) | 65 (65.6) | 26.2 (23.2–29.2) | 13.1 (12.8–13.4) | 1 (1.0) | 36.4 (36.3–36.4) |
| Chak | 100 | 32 (32.0) | 68 (68.0) | 26.8 (23.7–29.8) | 12.8 (12.5–13.1) | 0 (0.0) | 36.3 (36.2–36.3) |
| Chakma | 100 | 44 (44.0) | 56 (56.0) | 40.8 (37.6–44.0) | 13.1 (12.8–13.4) | 0 (0.0) | 36.2 (36.1–36.3) |
| Khumi | 100 | 45 (45.0) | 55 (55.0) | 31.3 (27.8–34.8) | 13.2 (12.9–13.5) | 0 (0.0) | 36.5 (36.4–36.6) |
| Khyang | 100 | 43 (43.0) | 57 (57.0) | 38.9 (35.2–42.6) | 12.8 (12.4–13.2) | 0 (0.0) | 36.1 (36.0–36.2) |
| *Lushai* | *2* | *1 (50.0)* | *1 (50.0)* | *50.5 (NA)* | *13.9** (13.1–14.7)* | *0 (0.0)* | *36.4 (32.9–39.9)* |
| Marma | 99 | 40 (40.4) | 59 (59.6) | 28.3 (25.0–31.6) | 12.2 (11.9–12.6) | 1 (1.0) | 36.1 (36.0–36.2) |
| Mro | 100 | 37 (37.0) | 63 (63.0) | 27.6 (24.1–31.1) | 13.7 (13.4–13.9) | 0 (0.0) | 36.1 (36.1–36.2) |
| Tangchangya | 100 | 49 (49.0) | 51 (51.0) | 28.5 (25.2–31.9) | 13.2 (12.9–13.6) | 1 (1.0) | 36.2 (36.1–36.3) |
| Tripura | 99 | 33 (33.3) | 66 (66.7) | 36.8 (33.1–40.5) | 12.6 (12.3–12.9) | 0 (0.0) | 36.3 (36.2–36.4) |
| **Total** | **999** | **401 (40.1)** | **598 (59.9)** | **32.5 (31.4–33.6)** | **13.0 (12.9–13.1)** | **4 (0.4)** | **36.2 (36.2–36.3)** |

                                    

The proportion of males within each ethnic group ranged from 40.1% to 59.9% and did not differ significantly (p = 0.279) (Table 1). Mean Hb concentration was 13.0 U/gHb (95% confidence interval (CI): 12.9 to 13.1, total range: 4.8 g/dL—19.3g/dL). A total of 11 participants were below the age of 7, the ratio of males to females did not differ between those above and below 7 years of age (p = 0.110). When comparing median G6PD activities between both cohorts, no significant difference was observed (p = 0.1418). The mean axillary body temperature was 36.3˚C (95% confidence interval (95%CI): 36.2–36.3) and 0.4% of all participants (4/ 1002) had a temperature above 37.5˚C (Table 1). Hb and body temperature differed significantly between ethnic groups, and this was also apparent after stratifying by sex (all p<0.001).

Three participants were diagnosed with *P. falciparum* by microscopy; all three cases were confirmed by RDT. Four microscopy-negative cases were also diagnosed as parasite-positive by RDT, including 3 *P. falciparum* cases and one participant diagnosed with a mixed *P. falciparum* and *P. vivax* infection; two of these participants (both *P. falciparum)* reported a history of malaria in the last 90 days. Malaria qPCR was undertaken successfully in all cases. Six of the seven malaria patients positive by RDT were confirmed by qPCR, one participant diagnosed with *P. falciparum* by RDT and microscopy returned a negative PCR result, and 36 participants were positive by PCR only. Based on PCR a total of 21 participants were positive for *P. falciparum*, 16 participants were positive for *P. vivax* and 5 participants harbored both *P. falciparum* and *P. vivax* parasites.

Relative to uninfected participants, the mean Hb concentration was significantly lower in patients diagnosed with malaria by microscopy 11.2g/dl (95%CI: 6.2 to 16.2) vs. 13.0 g/dl (5% CI: 12.9 to 13.1); p = 0.0264) or RDT (11.2 g/dL (95%CI: 9.9 to 12.5) vs. 13.0 (95%CI: 12.9 to 13.1); p = 0.0018) but not by the PCR result (p = 0.3285).

The median duration between sample collection and measurement of G6PD activity in the reference laboratory was 2 days (IQR: 0 to 4, range 0 to 7). 100% G6PD activity was defined as 7.03 U/gHb (interquartile range (IQR): 5.38–8.69, total range: 0.73–17.17) and median G6PD activity varied significantly with ethnicity when considering only males (p<0.001) or females (p<0.001). G6PD activities across the entire study population showed a bimodal distribution, which was more apparent in males than females (Fig 2).

The lowest AMM was detected among the Bengali sub-population (5.43 U/gHb, IQR: 4.78– 6.46), however, when females were included, the Chak had the lowest median G6PD activity (4.92 U/gHb, IQR: 1.83–7.12); Fig 3, Table 2, S1 Fig.

Median G6PD activity was 1.7U/gHb (24.2% of the AMM) higher among participants with peripheral malaria confirmed by microscopy or RDT compared to those with a negative result (8.6 U/gHb (IQR: 7.5 to 10.4) vs. 6.9 U/gHb (IQR: 5.2 to 8.6)), however the difference was not significant (p = 0.1894) and G6PD activities also did not differ between malaria and non-malaria participants when considering PCR instead (p = 0.6558) (Fig 4).

Overall, 90 (9.0%) participants were identified to be G6PD deficient (<30% enzyme activity), the prevalence being 12.7% (51/401) in males and 6.5% (39/598) in females; p = 0.001. A total of 133 (13.3%) individuals had intermediate G6PD activity based on the population wide AMM, which was present in 12.2% (49/401) of males and 14.0% (84/598) of females; p = 0.465. The proportions of individuals with G6PD deficiency differed significantly with ethnicity (p<0.001) with severe G6PD deficiency (below 30%) present in 26% (26/100) of participants of the Chak ethnicity but only 2.0% (2/99) of those of Tripura ethnicity; Table 2.

A total of 323 participants were genotyped (90 G6PD deficient individuals, 133 G6PD intermediate and 10 randomly selected G6PD normal participants per ethnic group (except Lushai); Table 3. Of the 223 participants diagnosed with G6PD activities below 70%, a G6PD variant was identified in 94 (42.2%) cases, comprising 68/90 (75.6%) G6PD deficient individuals and 26/133 (19.5%) intermediate deficient individuals. A total of 2 (2.0%) males and 10

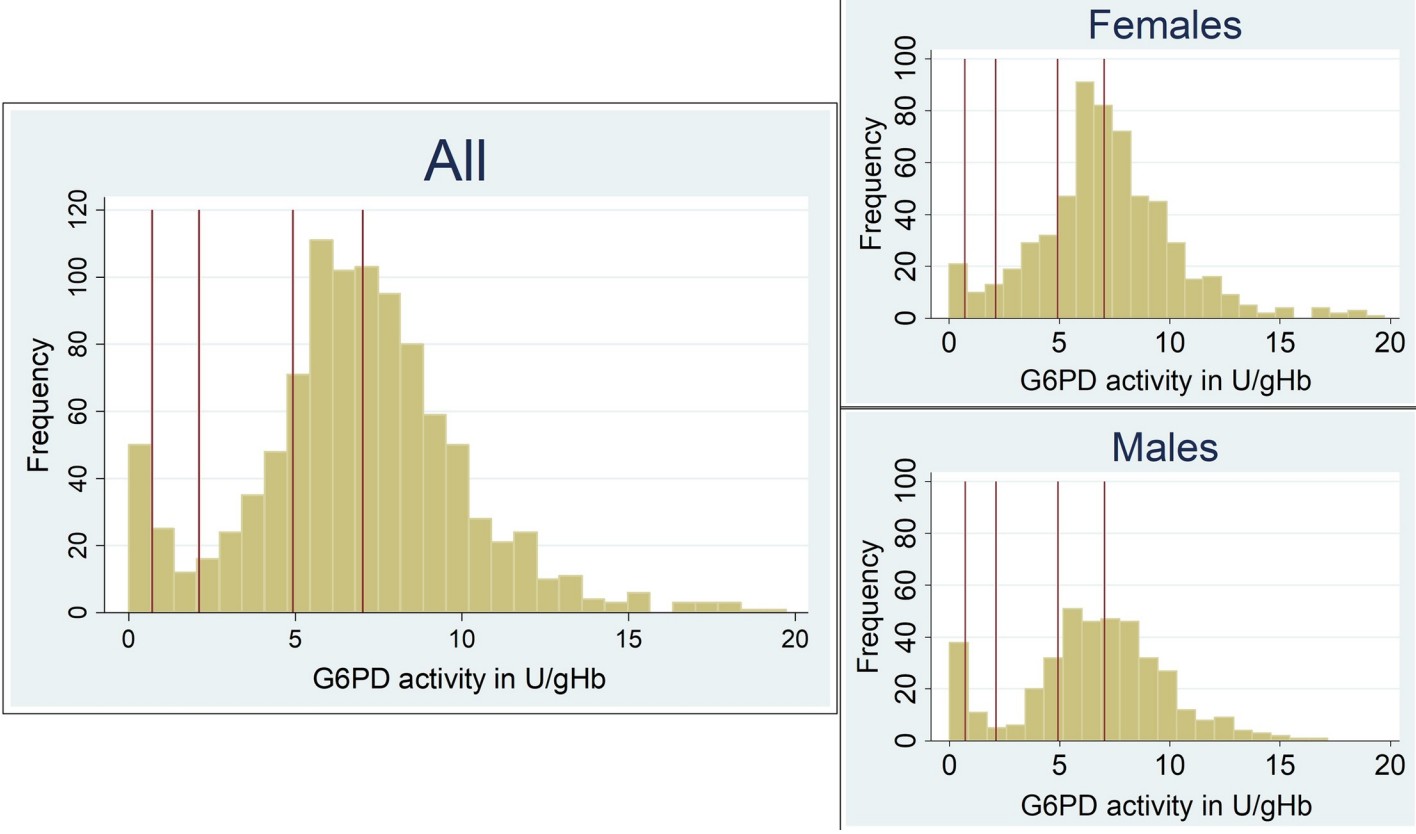

**Fig 2. Histogram of G6PD activities for the entire study population and stratified by sex.** Vertical lines from left to right indicate 10%, 30% 70% and 100% of the AMM.

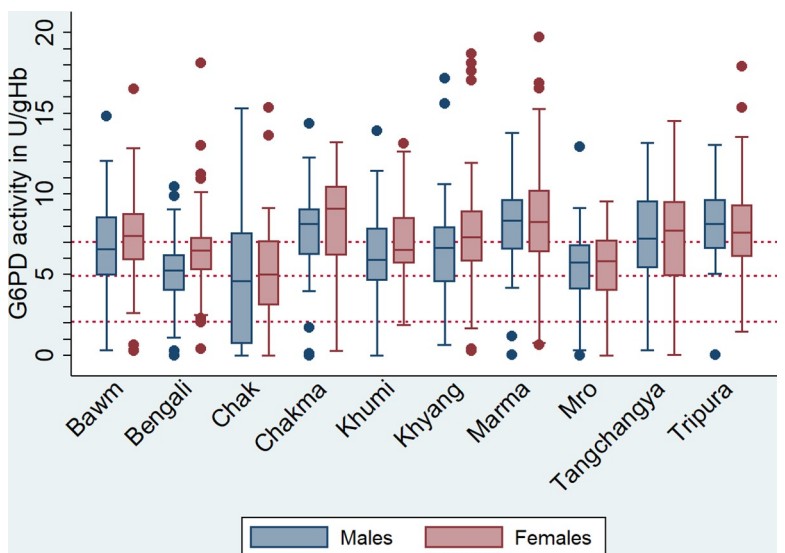

**Fig 3. Distribution of G6PD activity per ethnicity, stratified by sex.** Horizontal lines indicate 100%, 70%, and 30% G6PD activity of the AMM (from top to bottom).

**Table 2. Overview on G6PD activity and genotype per ethnicity.**

| Ethnicity | Total | AMM in U/gHb (IQR) | Number with <30% G6PD activity (%) | Number with >30%—<70% G6PD activity (%) | Number with hemizygous / homozygous Mahidol | Number with heterozygous Mahidol | Number with hemizygous / homozygous Orissa | Number with hemizygous Kalyan Kerala | Number with no variant determined (% of all genotyped) |
|---|---|---|---|---|---|---|---|---|---|
| Bawm | 100 | 7.12 (5.48–8.76) | 8 (8.0) | 9 (9.0) | 5 | 1 | 0 | 0 | 21 (77.8) |
| Bengali | 99 | 5.43 (4.78–6.46) | 6 (6.1) | 21 (21.2) | 0 | 2 | 2 | 1 | 32 (86.5) |
| Chak | 100 | 5.71 (1.27–7.93) | 26 (26.0) | 24 (24.0) | 20 | 20 | 0 | 0 | 20 (33.3) |
| Chakma | 100 | 8.36 (6.84–9.19) | 7 (7.0) | 5 (5.0) | 4 | 2 | 0 | 0 | 16 (72.7) |
| Khumi | 100 | 6.33 (5.04–7.98) | 7 (7.0) | 14 (14.0) | 5 | 4 | 0 | 0 | 22 (71.0) |
| Khyang | 100 | 6.70 (4.60–7.94) | 6 (6.0) | 15 (15.0) | 2 | 0 | 4 | 0 | 25 (80.7) |
| *Lushai* | *2* | *6.43 (0.00 to 27.52)\** | *0 (0.0)* | *1 (50.0)* | *0* | *0* | *0* | *0* | *1 (NA)* |
| Marma | 99 | 8.59 (6.78–9.64) | 7 (7.1) | 5 (5.1) | 4 | 1 | 0 | 0 | 17 (77.3) |
| Mro | 100 | 5.94 (4.38–7.23) | 11 (11.0) | 23 (23.0) | 6 | 9 | 0 | 0 | 29 (65.9) |
| Tangchangya | 100 | 7.30 (5.65–9.77) | 10 (10.0) | 12 (12.0) | 9 | 3 | 0 | 0 | 20 (62.5) |
| Tripura | 99 | 8.18 (6.65–9.77) | 2 (2.0) | 4 (4.0) | 2 | 0 | 0 | 0 | 14 (87.5) |
| **Total** | **999** | **7.03 (5.38–8.69)** | **90 (9.0)** | **133 (13.3)** | **57** | **42** | **6** | **1** | **217 (67.2)** |

\*arithmetic mean (95%CI)

females (10%) of the randomly selected G6PD normal participants had a detectable *G6PD* variant. Overall, the Mahidol variant was detected in 99 individuals (93.4% of all identified variants), the Orissa variant in 6 individuals (5.7% of all identified variants) and the Kalyan—Kerala variant in 1 individual (0.9% of all identified variants).

In total, 45 males were hemizygous and 12 females homozygous for the Mahidol variant with a median G6PD activity of 0.49 U/gHb (7.0% of the AMM, range: 0.0% to 185.8%); two males and one female had G6PD activities above 30%. In the 42 females identified as being heterozygous for the Mahidol variant, their median G6PD activity was 3.72 U/gHb (52.9% of the AMM, range: 0.0% to 206.5%). 11/42 (26.2%) were G6PD deficient (activity <30%), 21/42 (50.0%) intermediately deficient, and 10/42 (23.8%) had normal G6PD activity (Table 3).

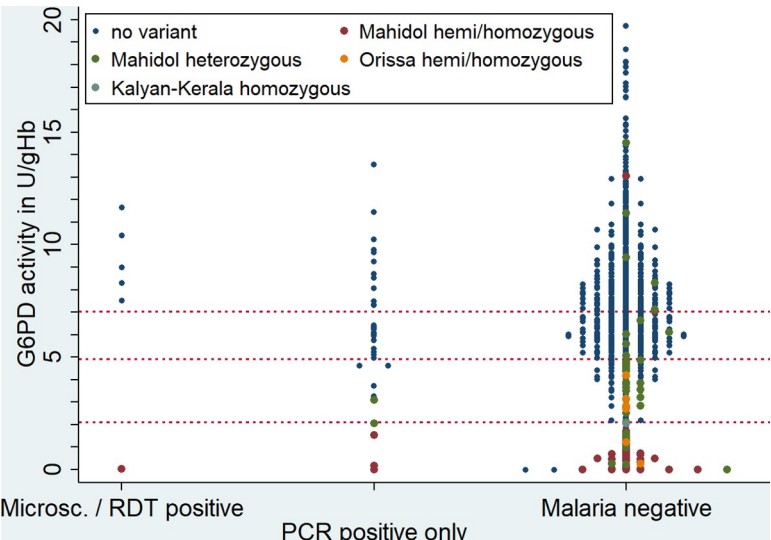

**Fig 4.** Distribution of G6PD activity and confirmed genotype per malaria diagnosis* Horizontal lines indicate 100%, 70%, and 30% G6PD activity of the AMM (from top to bottom) *one participant, positive by RDT and microscopy but negative by PCR was excluded.

The median G6PD activity among the 3 males hemizygous and 3 females homozygous for the Orissa variant was 2.77U/gHb (39.4% of the AMM; range 3.8%-59.6%) and 2.07 U/gHb (29.4% of the AMM) in one male hemizygous for the Kalyan-Kerala variant (Fig 5).

A multivariate regression model was generated to predict non-normalized G6PD activity (F $(17, 981)$ = 73.12, p<0.001, adjusted $R^2$ = 0.5513). Significant independent variables in order of contribution that predicted G6PD activity were genotype (except for the Kalyan Kerala variant), Hb concentration, ethnicity, sex and weight. Malaria diagnosis, irrespective of method, was not a significant covariate (S3 Table). There was no significant multi-co-linearity in either the initial or final models.

## Discussion

In the first study of G6PDd in the multi-ethnic CHT population, our results revealed that almost 10% of the population had G6PD activities of less than 30%, however the prevalence differed significantly between ethnic groups. Among identified *G6PD* variants the Mahidol variant accounted for 93% of all cases, similar to earlier reports from the area.

Differences in the prevalence of G6PDd among different ethnicities living in the same area and sharing the same risk of malaria infection had been reported earlier from other sites, however not to the same extent as reported here. PQ based radical cure is contraindicated in

**Table 3. Genotyping results per category of G6PD activity measured by spectrophotometry.**

| Genotype | G6PD activity <30% | G6PD activity ≥30% to <70% | G6PD activity ≥70% | Total |
|---|---|---|---|---|
| Total with no variant determined (% per row) | 22 (10.1) | 107 (49.3) | 88 (40.6) | 217 |
| Total hemi / homozygous Mahidol (% per row) | 54 (94.7) | 1 (3.8) | 2 (3.5) | 57 |
| Total heterozygous Mahidol (% per row) | 11 (26.2) | 21 (50.0) | 10 (23.8) | 42 |
| Total hemi / homozygous Orissa (% per row) | 2 (33.3) | 4 (66.7) | 0 (0.0) | 6 |
| Total homozygous Kerala (% per row) | 1 (100.0) | 0 (0.0) | 0 (0.0) | 1 |
| **Total** | 90 | 133 | 100 | **323** |

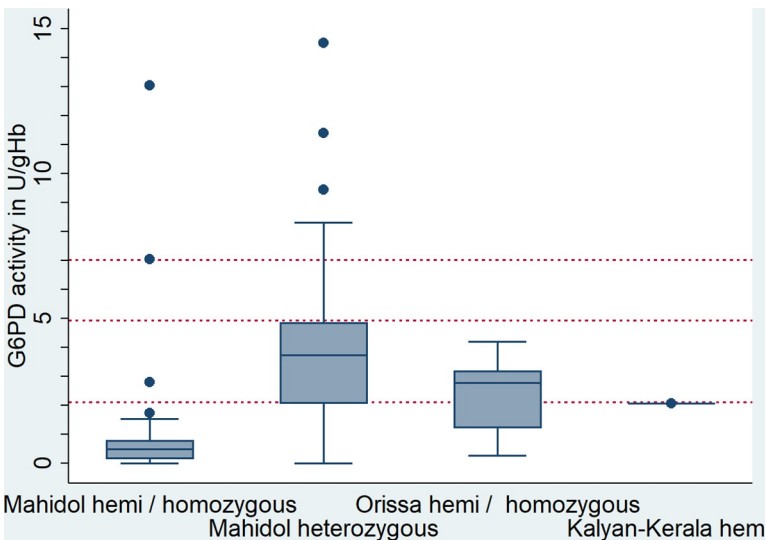

**Fig 5.** Distribution of G6PD activity per genotype. Horizontal lines indicate 100%, 70%, and 30% G6PD activity of the AMM (from top to bottom).

patients presenting with G6PD enzyme activity of less than 30% [16]. Using this criterion, only 74% of Chak would be eligible for treatment, compared to 98% of those from the Tripura tribe. Despite a high prevalence of G6PD deficiency and thus associated risk of drug induced hemolysis, there are few reports of PQ induced hemolytic reactions either from the local medical staff (C.S. Phru, personal communication) or in the literature [8,18]. The reasons for this are unclear, but could reflect low prescription of PQ by healthcare providers, underreporting of adverse events due to limited access to healthcare, poor treatment adherence to a prolonged course of PQ, or G6PDd individuals having partial protection against malaria [15,41,42]. In earlier hospital-based malaria surveys conducted in the same area, less than 2% of malaria patients were G6PDd, considerably less than the 9% prevalence we found within the mostly healthy population of this survey. While not statistically significant, the median G6PD activity was almost 25% higher (1.7U/gHb) among patients with malaria diagnosed by microscopy or RDT compared to participants with a negative result. Malaria status was not a significant predictor of G6PD activity irrespective of diagnostic assay applied, however our study was neither designed nor powered to this effect.

Although G6PD genotype is a key determinant of enzyme activity [43], we observed poor correlation between G6PD genotype and phenotype, a phenomenon that reflect both assay variability as well as host and environmental factors [44–49]. Almost 25% of females heterozygous for a known local G6PD variant had activities above 70%, likely due to lyonization patterns in favor of the G6PD normal allele [50]. Interestingly, 5% of participants, hemi- or homozygous for the Mahidol variant, that confers very low G6PD levels, had either intermediate or normal G6PD activities. Although not quantified, study participants may have had recent blood loss due to injury, or substance or pathogen induced hemolysis, thereby replacing older erythrocytes with reduced G6PD activity with younger RBCs with higher activity [51]. Neither can we exclude the possibility that concomitant hemoglobinopathies or high leukocyte counts, may have also confounded measured G6PD activity [37].

The study had some inherent limitations. The panel of five *G6PD* variants that were genotyped was based on earlier reports from the area [8,26,52]. However, a known *G6PD* variant was only present in 42% of patients with less than 70% G6PD activity. Given that none of the

samples failed PCR, it is likely that the chosen panel did not include all variants present in the region. And even in those in whom a *G6PD* variant was identified, the correlation between genotype and phenotype was not good, suggesting either erroneous results from spectropho- tometry, possibly arising from delays in sample transport to the reference laboratory or proce- dural errors. However, all samples were tested within seven days, the recommended maximum interval between venipuncture and measurement of G6PD activity [33]. Furthermore, stan- dardized commercial controls were run with every measurement, all samples were tested in duplicate, and results only accepted if both measurements did not differ by more than 10%.

In conclusion, there was a high prevalence of G6PDd in the CHT, but this varied signifi- cantly between ethnic groups. Whilst testing should ideally be made widely available to identify individuals with G6PDd, in practice this is often unavailable or unreliable due to financial and logistical constraints. Priority should be given to ensuring the availability of G6PDd testing in communities where the prevalence of G6PDd and thus hemolytic risk is greatest. In areas where testing is not available, consideration should be given to administering primaquine with additional measures to mitigate severe adverse reactions by detecting early signs of hemolysis. Determining the prevalence and variants of G6PDd in malaria endemic areas with different ethnic groups will inform public health interventions to ensure that primaquine radical cure can be provided safely and effectively.

## Supporting information

**S1 Table. Primer sequences for polymerase chain reactions.**
(DOCX)

**S2 Table. Annealing temperatures, post PCR product base pair and restriction enzymes for the PCR-RFLP protocol.** *Optimized temperature.
(DOCX)

**S3 Table. Multivariate regression model to predict non-normalized G6PD activity (n = 999, F17, 981) = 68.27, p<0.001, adjusted R$^2$ = 0.5513).**
(DOCX)

**S1 File. Village names / ethnicity.**
(XLSX)

**S2 File. Underlying database.**
(CSV)

**S1 Fig. G6PD activity per ethnicity. Horizontal lines indicate 100%, 70%, and 30% G6PD activity of the AMM (from top to bottom).**
(TIF)

## Acknowledgments

We would like to thank all participants for their time and support as well as all staff that has made this survey possible. icddr,b is grateful to the Governments of Bangladesh, Canada, Swe- den and the UK for providing core/unrestricted support.

## Author Contributions

**Conceptualization:** Benedikt Ley.

**Data curation:** Mohammad Sharif Hossain.

**Formal analysis:** Benedikt Ley.

**Funding acquisition:** Ric N. Price.

**Investigation:** Ching Swe Phru.

**Methodology:** Mohammad Golam Kibria, Nusrat Jahan, Fatema Tuj Johora, Jenifar Quaiyum Ami, Cristian Koepfli.

**Project administration:** Ching Swe Phru.

**Supervision:** Wasif Ali Khan, Mohammad Shafiul Alam.

**Writing – original draft:** Benedikt Ley.

**Writing – review & editing:** Benedikt Ley, Sarah Auburn, Kamala Thriemer, Ric N. Price, Cristian Koepfli, Mohammad Shafiul Alam.

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
