## [Decision Letter · Decision Letter 0]

28 May 2020

Dear Dr Ley,

Thank you very much for submitting your manuscript "Wide range of G6PD activities found among ethnic groups of the Chittagong Hill Tracts, Bangladesh" for consideration at PLOS Neglected Tropical Diseases. As with all papers reviewed by the journal, your manuscript was reviewed by members of the editorial board and by several independent reviewers. In light of the reviews (below this email), we would like to invite the resubmission of a significantly-revised version that takes into account the reviewers' comments. 

The manuscript "Wide range of G6PD activities found among ethnic groups of the Chittagong Hill Tracts, Bangladesh" was evaluated by 3 reviewers who issued the opinions below.

Assessment of phenotypes for G6PD is really valuable. Unfortunately, the quality of the data as presented so far adds limitations to the manuscript. The limitations need to be discussed to avoid these findings might be then generalized and used inappropriately in the future. To be even more clear, the idea that there is not good correlation between phenotypes and genotypes in G6PD is most often caused by poor laboratory technique or poor sample handling. Technical limitations of the manuscript need to be highlighted.

I recommend to present full data set in an universal format such as csv.

We cannot make any decision about publication until we have seen the revised manuscript and your response to the reviewers' comments. Your revised manuscript is also likely to be sent to reviewers for further evaluation.

Sincerely,

Wuelton Marcelo Monteiro, Ph.D.

Associate Editor

Mary Lopez-Perez

Deputy Editor

The manuscript "Wide range of G6PD activities found among ethnic groups of the Chittagong Hill Tracts, Bangladesh" was evaluated by 3 reviewers who issued the opinions below.

Assessment of phenotypes for G6PD is really valuable. Unfortunately, the quality of the data as presented so far adds limitations to the manuscript. The limitations need to be discussed to avoid these findings might be then generalized and used inappropriately in the future. To be even more clear, the idea that there is not good correlation between phenotypes and genotypes in G6PD is most often caused by poor laboratory technique or poor sample handling. Technical limitations of the manuscript need to be highlighted.

I recommend to present full data set in an universal format such as csv.

Reviewer's Responses to Questions

**Key Review Criteria Required for Acceptance?**

**Methods**

-Are the objectives of the study clearly articulated with a clear testable hypothesis stated?

-Is the study design appropriate to address the stated objectives?

-Is the population clearly described and appropriate for the hypothesis being tested?

-Is the sample size sufficient to ensure adequate power to address the hypothesis being tested?

-Were correct statistical analysis used to support conclusions?

-Are there concerns about ethical or regulatory requirements being met?

Reviewer #1: Describe the selection of participants for genotyping in the methods section?

Reviewer #2: Yes (please see comments below)

Reviewer #3: (No Response)

**Results**

-Does the analysis presented match the analysis plan?

-Are the results clearly and completely presented?

-Are the figures (Tables, Images) of sufficient quality for clarity?

Reviewer #1: “… Hill Tracts (CHT) carry the highest malaria burden …” better say people living in the Hill tracts?

Table 1: the column “mean body temp. …” is not helpful and could be omitted or replaced with number of febrile participants detected?”. Same is true for the sentence “The mean axillary body temperature was 36.3°C (95% confidence interval (95%CI): 36.2 – 224 36.3) …”

 A figure illustrating the assembly of the participants (consort chart) could be useful?

The regression model is difficult to understand. Please revise the Table 5 so as to show the affected/non affected number of participants for each row in a separate column? 

There is something missing in the sentence: “…almost 10% of the population of the had G6PD activities of less than 30%...”

Reviewer #2: Partly please see comments below)

Reviewer #3: (No Response)

**Conclusions**

-Are the conclusions supported by the data presented?

-Are the limitations of analysis clearly described?

-Do the authors discuss how these data can be helpful to advance our understanding of the topic under study?

-Is public health relevance addressed?

Reviewer #1: The authors speculate regarding the low number of disease/symptom presented by G6PD deficient individuals. “Despite a high proportion of the population being at risk of drug induced hemolysis, there are few reports of drug induced hemolytic reactions from local medical staff (B. Ley, personal communication) and the literature”. This requires more explanations. How are these deficient individuals challenged? Do they receive routinely PQ treatment or other triggers for haemolysis? In the absence of information on exposure to triggers for haemolysis it is difficult to assess the outcome.

I am not convinced by the conclusion that testing could/should be conducted according to ethnicity as even in tribes with low G6PD def. prevalence deficient individuals were detected. It may be wiser to recommend routine use of rapid tests/biosensors for wider use of 8-aminoquinoline regimens?

Reviewer #2: Partly please see comments below)

Reviewer #3: (No Response)

**Editorial and Data Presentation Modifications?**

Reviewer #1: not applicable...

Reviewer #2: (No Response)

Reviewer #3: (No Response)

**Summary and General Comments**

Reviewer #1: A very nicely designed study which provides new information on G6PD deficiency essential for the rollout of the radical cure and hence the elimination of vivax malaria.

Reviewer #2: The study by Ley et al. is a well-conducted study about G6PDd prevalence in Bangladesh population. Both genotype and phenotype were screened for subjects from 11 different ethnicities. The novelty is that Ley´s study brings a more comprehensive picture regarding G6PD activity in that population. This is important in the context of radical cure of P. vivax malaria. Whilst some of the data are interesting, the authors need to clarify many points before publication. 

Major comments

1. The authors show the mean Hb values for each population in Table 1; however, this data should be presented and analyzed stratified by gender since it differs between men and women.

2. It is not clear why authors included the two individuals of Lushai ethnic group in the analysis since only two subjects are not representative. I suggest removing them and consider analyzing 10 ethnic groups. 

3. From the main text or tables I could not find the results for qPCR concerning parasite species. Please add this information to the manuscript.

4. It is clear from previous studies that heterozygous women cannot be accurately identified through G6PD enzyme activity assays. I suggest presenting G6PD activity data in Table 2 and Fig. 2 stratified by gender. Did the authors find significant differences in G6PD activity among ethnic groups stratified by gender? This could be due to different populations presenting different variants of G6PD gene (for variants assayed and for those that could not be assayed in the present study).

5. In the Discussion, the authors claim that malaria status was not found to be a significant predictor of G6PD activity irrespective of diagnostic assay applied. However, it should be made clear that the study was not powered to assess any association between G6PDd and positivity for malaria.

6. In Lines 339-342, it was raised the possibility of a recent haemolytic event to explain inconsistent results for genotype-phenotype correlations. But other factors as concomitant haemoglobinopathies and high leukocyte counts could also lead to a false normal G6PD result (Malaria Journal, 2013; 12:391). Did the authors take these other issues into account? Please, discuss these potential factors. 

7. The authors should evaluate the possibility (due to sample size) of estimating the Hardy-Weinberg equilibrium for each ethnic group.

Minor comments

Line 161. Please add the reference of qPCR assays used for molecular diagnosis of malaria. 

Lines 166-167. Specify the final concentration of each reagent (or the volume used and the stock concentration) used in the PCR reaction.

Line 219. This data was not shown in Table 1. This information needs to be removed.

Line 305. Multivariate is more appropriate than “multivariable” (please see in Am J Public Health., 2013; 103(1): 39–40). While multivariable analysis refers to statistical models in which there are multiple response variables, multivariate analysis refers to statistical models that have ≥ 2 outcome variables.

Table 2. I suggest including the number of individuals with no variant determined.

Table 2. Correct “Number with with”.

Reviewer #3: This is an interesting manuscript that describes G6PD genotypes and phenotypes in a malaria endemic area of Bangladesh where potentially hemolytic radical cure for Plasmodium vivax will be used. The aim of the study is reasonable but the laboratory analyses have major issues and some of the results interpretation are questionable. The description of the analyzed population and the sample handling is also not very detailed. Probably my biggest question is: why the authors have used the Hb value assessed in the field (up to 7 days earlier) instead of repeating the Hb measurement the same day as spectrophotometric test? This would have been really easy (with hemocue or CBC) and would have given much more reliable results.

Major issues

Line 123 Can the authors show a map with villages and ethnicities? Was the delay from sample collection to analysis in central lab correlated to village of sampling and/or ethnicity?

Line 140 participants <7, how many? Which age range? It is known that G6PD activity is higher in the first 6 months of life, so if small babies were included in the study it should be clearly stated.

Line 155 procedures described earlier need a reference and more info on number of replicates and CV among them, temperature conditions, Quality Controls, accuracy, etc. As far as I have seen in literature, the Randox system does not seem extremely accurate for G6PD activity so the authors might want to give more details to show how it performs in their setting.

Line 156 Hb from the field could be potentially highly unreliable if sample have been shipped without appropriate control of temperature, they could have hemolysed and Hb estimation changes when samples get older (usually after 2-3 days).

Line 192 “The dependent variable was non-normalized G6PD activity (in U/dL), independent variables were backwards selected, initially considering age, gender, body weight, height and temperature, Hb, the delay between sample collection and processing, spectrophotometry result, and G6PD genotype”. Please explain why you used the non-normalized G6PD activity, how gender is independent from G6PD and why body weight and height should be considered variables in this analysis. Please explain why this analysis is even needed.

Line 203 how were the 10 sample per ethnic group chosen? Where they both females and males? Where they from different villages?

Line 239 The delay between sample collection and analysis is rather large; what is written here does not correspond to what written in the discussion in Line 349. 

Line 240 “100% G6PD activity was defined as 7.03 U/gHb (interquartile range (IQR): 5.38 – 8.69, total range: 0.73 – 17.17) and median G6PD activity varied significantly with ethnicity (p<0.001).“ Is this the AMM or the Male Median (MM)? The median activity between groups should be compared within the same gender (especially since gender were not represented equally in all groups).

Line 249 “when females were included, the Chak had the lowest median G6PD activity (4.92 U/gHb,

 IQR: 1.83 – 7.12)”; unclear what this means. The AMM in the Chak (Table 2) has a very low first quartile, is this really the AMM or the MM?

Line 257-262 and Table 3: these comparisons can only be made if the proportion of females and males in the different groups is the same.

Line 274: “A total of 133 (13.3%) individuals had intermediate G6PD activity, which was present in 12.2% (49/401) of males and 14.0% (84/598) of females; p=0.465.” The authors need to explain how this is possible. The expected number of males with intermediate activity in a population should be very little. Are these data calculated using the overall AMM or the ethnic specific AMM?

Line 281-285 “Of the 223 participants diagnosed with G6PD activities below 70%, a G6PD variant was identified in 94 (42.2%) cases, comprising 68/90 (75.6%) G6PD deficient individuals […]. A total of 12 (12.0%) of the randomly selected G6PD normal participants had a detectable G6PD variant.” I see two problems with results here: 1) 75.6% of deficient individuals had none of the analysed mutations. While this is possible, the authors should bring some comparative data on prevalence of other known mutations in the area (not included in this analysis) to give us an idea of how likely it is that they make up >25% of deficient samples. 2) 12% of G6PD normal subjects had a variant, if they are females there is no problem at all, if they are males this needs to be explained.

Line 288 and Table 4 “In total, 45 males were hemizygous and 12 females homozygous for the Mahidol variant with a median G6PD activity of 0.49 U/gHb (7.0% of the AMM, range: 0.0% to 185.8%)”; same here, based on published data, a hemi or homozygous Mahidol is not expected to have >50% activity, is there any plausible explanation for this result?

Line 305-312 I do not see the point of using a regression model and in particular to use the non-normalized G6PD activity.

Line 339 Although this is a possible explanation, I would like the authors to also bring other more likely explanations for this large proportion of genetically deficient but phenotypically normal subjects. As mentioned before, analysing the Hb in the field few days prior the assessment of G6PD activity might have brought a large variation in the G6PD activities. A test such as Hemocue (or even CBC) would have been pretty easy to run in the equipped central laboratory at the same time as G6PD. The delay in G6PD analysis, the way samples were transported (was the temperature checked during transportation? Did any sample show signs of hemolysis?), whether the sample came from small children or adults, accuracy of laboratory test used, village of origin or sampling day of samples with unexpected G6PD phenotypes…all these are factors that need to be clearly stated in the manuscript and taken into consideration when analysing the results.

Line 355 “Testing should be considered at least for ethnic groups with high proportions of G6PDd and G6PD intermediate individuals, albeit ethnicity specific testing may be difficult to implement in practice.” Since the data show that all ethnic groups analysed, with the exception of “Tripura”, have a prevalence of G6PD deficiency >5%, it seems that a universal G6PD testing strategy would be more appropriate than an ethnic specific one.

Minor issues/comments

Line 36 by reading the methods, it is clear that participants where not randomly enrolled. This should be corrected in the whole manuscript.

Line 146 50% discrepancy seems pretty high

Line 164 how was the subset chosen?

Line 187 why you test for normality of G6PD activity distribution which is expected to be bimodal?

Line 214 suggest eliminate the 2 samples from the Lushai ethnic group

Line 224 “While Hb and body temperature differed significantly among ethnic groups (all p<0.001)”; was sex and age taken into account?

Line 325-334 The authors should give some info on how many subjects get actually treated with long course primaquine in the region; low access to primaquine could completely justify the lack of haemolytic reactions.

Line 331 if it is not significant, it is not higher! I also do not understand this sentence “malaria status was not found to be a significant predictor of G6PD activity irrespective of diagnostic assay applied.”

Line 335 how many papers instead do report a good correlation between G6PD genotype and phenotypes? 

Figure1 shows that the distribution in males, although bimodal, has a large number of subjects in the intermediate range

Figure 2 I would show only males

PLOS authors have the option to publish the peer review history of their article (what does this mean?). If published, this will include your full peer review and any attached files.

Reviewer #1: No

Reviewer #2: No

Reviewer #3: No
---

## [Decision Letter · Decision Letter 1]

8 Jul 2020

Dear Dr. Ley,

Thank you very much for submitting your manuscript "Wide range of G6PD activities found among ethnic groups of the Chittagong Hill Tracts, Bangladesh" for consideration at PLOS Neglected Tropical Diseases. As with all papers reviewed by the journal, your manuscript was reviewed by members of the editorial board and by several independent reviewers. The reviewers appreciated the attention to an important topic. Based on the reviews, we are likely to accept this manuscript for publication, providing that you modify the manuscript according to the review recommendations. 

Sincerely,

Wuelton Marcelo Monteiro, Ph.D.

Associate Editor

Mary Lopez-Perez

Deputy Editor

Reviewer's Responses to Questions

**Key Review Criteria Required for Acceptance?**

**Methods**

-Are the objectives of the study clearly articulated with a clear testable hypothesis stated?

-Is the study design appropriate to address the stated objectives?

-Is the population clearly described and appropriate for the hypothesis being tested?

-Is the sample size sufficient to ensure adequate power to address the hypothesis being tested?

-Were correct statistical analysis used to support conclusions?

-Are there concerns about ethical or regulatory requirements being met?

Reviewer #1: (No Response)

Reviewer #2: (No Response)

Reviewer #3: (No Response)

**Results**

-Does the analysis presented match the analysis plan?

-Are the results clearly and completely presented?

-Are the figures (Tables, Images) of sufficient quality for clarity?

Reviewer #1: (No Response)

Reviewer #2: (No Response)

Reviewer #3: (No Response)

**Conclusions**

-Are the conclusions supported by the data presented?

-Are the limitations of analysis clearly described?

-Do the authors discuss how these data can be helpful to advance our understanding of the topic under study?

-Is public health relevance addressed?

Reviewer #1: (No Response)

Reviewer #2: (No Response)

Reviewer #3: (No Response)

**Editorial and Data Presentation Modifications?**

Reviewer #1: (No Response)

Reviewer #2: (No Response)

Reviewer #3: (No Response)

**Summary and General Comments**

Reviewer #1: (No Response)

Reviewer #2: The authors have satisfactorily responded to all my questions and made the necessary changes to the manuscript.

Reviewer #3: The authors have adequately addressed some of my questions but there are a number of issues that have not been resolved and some have arisen during revision. In general, I feel the authors have not explored all the aspects of the questions and comments received from the reviewers. 

They state their main aim in line 34 and again line 102: “Aim of this study was to determine the need for routine testing within a multi-ethnic population by assessing the prevalence of G6PDd among the local population”. If the aim was to assess the prevalence of G6PD deficiency, a qualitative test (in the field or in the lab) would have been sufficient. The authors took a brave decision to perform a rather more complicated work in order to bring more details and possibly useful data but then fail to perform the reference test in the correct way (using Hb from previous assessment) and in their main statistical analysis they do not use the right values of G6PD activity (see my comment below). After showing that all ethnic groups have a moderate to high prevalence of G6PD deficiency they reach the conclusion that “practically” G6PD screening might still be done based on ethnicity (even after 3 reviewers suggested that was not the right interpretation).

And while it is useful to know that the laboratory performing the analysis collaborates with WHO and PATH, I believe the comments and suggestions from reviewers where certainly not about the quality or reputation of the laboratory but about the quality of data presented, possibly the study design and quite clearly the conclusions. 

Below there are few more comments to specific issues:

1) In my first review I had asked about the accuracy of Randox G6PD assessment and to specify whether testing was carried out in single replicate or duplicate. Accuracy is not reported and there is no mention of duplicates in the methods but there is a mention in line 359-361 of discussion. This should be written in the methods. It is also unclear to me why the haemoglobin was not re-assessed at the laboratory when the samples were analysed for G6PD, which is in my opinion one of the biggest technical limitations of this study. The authors replied that in previous studies they have tested samples kept in laboratory for up to 1 week but I presume they had retested both G6PD and Hb over time. 

2) In the multivariate analysis, the variable listed in the methods (line 199-202) are not the same as those listed in the results (315-317). For example, “ethnicity” is missing in the variable listed in methods. And on a more substantial aspect of this analysis:

a) I think the authors are wrong in using the “non-normalized” values of absorbance, because G6PD activity is defined as normalized by either Hb or number of RBCs. The “non-normalized” absorbance is simply not the enzymatic activity.

b) While they make a point of not using Hb-normalized G6PD in order to then analyse Hb as independent variable, they use a lot of variables that are actually collinear (sex and heterozygote genotype, sex and weight, sex and height) and others that might be collinear (for example if a mono-ethnic village was very far away and samples analysis was delayed, then “ethnicity” and “delay to test” variables would be collinear). I believe this should be checked and made clear in the analysis. 

3) Lushai ethnic group: a sentence in the text would be enough to report that G6PD deficiency was found in 1 person of this ethnicity, without needing to add this ethnic group in the table. 

4) The authors confirm that “Although G6PD genotype is a key determinant of enzyme activity, we observed poor correlation between G6PD genotype and phenotype, a phenomenon that has been described previously (44-47).” The first paper used retrospective hospital data collected during a decade (1993-2003) and phenotype were defined using four different ranges over time; hardly comparable to the current study. The second paper shows similar poor genotype-phenotype correlation and indeed used the same reagents as the current one. I think there are many reasons to believe that the poor correlation observed here and rarely in other settings is due primarily to technical issues rather than reflecting a true biological phenomenon.

PLOS authors have the option to publish the peer review history of their article (what does this mean?). If published, this will include your full peer review and any attached files.

Reviewer #1: No

Reviewer #2: No

Reviewer #3: No
---

## [Editor Report · Decision Letter 2]

11 Aug 2020

Dear Dr. Ley,

We are pleased to inform you that your manuscript 'Wide range of G6PD activities found among ethnic groups of the Chittagong Hill Tracts, Bangladesh' has been provisionally accepted for publication in PLOS Neglected Tropical Diseases.

Best regards,

Wuelton Marcelo Monteiro, Ph.D.

Associate Editor

Mary Lopez-Perez

Deputy Editor

After the return of the modified manuscript, I believe that all requests were met properly. I suggest accepting the manuscript as it is in its latest version.

---

## [Editor Report · Acceptance letter]

2 Sep 2020

Dear Dr Ley,

We are delighted to inform you that your manuscript, "Wide range of G6PD activities found among ethnic groups of the Chittagong Hill Tracts, Bangladesh," has been formally accepted for publication in PLOS Neglected Tropical Diseases.

Best regards,

Shaden Kamhawi

co-Editor-in-Chief

Paul Brindley

co-Editor-in-Chief
